# POSQA: Probe the World Models of LLMs with Size Comparisons

**Chang Shu\***
University of Cambridge
cs2175@cam.ac.uk

**Jiuzhou Han\***
Monash University
jiuzhou.han@monash.edu

**Fangyu Liu**[†]
Google DeepMind
liufangyu@google.com

**Ehsan Shareghi**
Monash University
ehsan.shareghi@monash.edu

**Nigel Collier**
University of Cambridge
nhc30@cam.ac.uk

## Abstract

Embodied language comprehension emphasises that language understanding is not only a matter of mental processing in the brain, but also involves interactions with the physical and social environment. With the explosive growth of Large Language Models (LLMs) and their already ubiquitous presence in our daily lives, it is becoming increasingly necessary to verify their real-world understanding. Inspired by cognitive theories, we propose **POSQA**: a **P**hysical **O**bject **S**ize **Q**uestion-**A**nswering dataset with simple size comparison questions to examine the extremity and analyse the potential mechanisms of the embodied comprehension of the latest LLM.

We show that even the largest LLMs today perform poorly under the zero-shot setting. We then push their limits with advanced prompting techniques and external knowledge augmentation. Furthermore, we investigate whether their real-world comprehension primarily derives from contextual information or internal weights and analyse the impact of prompt formats and report bias of different objects. Our results show that real-world understanding that LLMs shaped from textual data can be vulnerable to deception and confusion by the surface form of prompts, which makes it less aligned with human behaviours.

## 1 Introduction

The rapid growth of recent Large Language Models (LLMs) such as ChatGPT has led to their increased use in various applications (Gozalo-Brizuela and Garrido-Merchán, 2023; Sobania et al., 2023; Lehnert, 2023; Guo et al., 2023; Nov et al., 2023; Jiao et al., 2023). With the rapid growth of interest in developing Embodied Language Models (ELM) (Dasgupta et al., 2023; Driess et al., 2023; Vemprala et al., 2023), recently there has been increas-

ing interest in investigating whether LLM have an aligned understanding of the real world as our human from cognitive and physiological perspectives (Prystawski et al., 2022; Binz and Schulz, 2022; Hagendorff et al., 2022; Mahowald et al., 2023). Embodied language comprehension (Horchak et al., 2014; Buccino et al., 2016; Fischer and Zwaan, 2008; Barsalou, 1999), a possible explanation for human cognition, suggests that the human develops an understanding of the physical world related by language by our physical experiences and sensory perceptions of the world around us. When we process languages, we reemulate or recreate the experiences mentioned in the language to understand and interact with those languages more meaningfully.

Although common sense physical reasoning has been widely explored previously with various benchmarks, such as PIQA (Bisk et al., 2020b), MMLU-Physics (Hendrycks et al., 2021), UTOPIA (Liu et al., 2022b), and PROST (Aroca-Ouellette et al., 2021), few studies analyse the understanding of LLMs about object size, which is actually central to various fundamental aspects of cognition such as implicit memory, object recognition, conceptual processing, and perception-action coordination (Biederman and Cooper, 1992; Barsalou, 2008). Therefore, inspired by cognitive experiments (de Koning et al., 2017a), we proposed **POSQA**: a **P**hysical **O**bject **S**ize **Q**uestion-**A**nswering dataset containing 12,000 questions of size comparisons between pairs of objects to investigate whether the latest LLMs have aligned cognition with our human and identify the limits of their real-world understanding with various prompt-based experiments.

Empirical findings suggest that under the zero-shot setting, the performance of popular LLMs such as GPT-3 is slightly better than random guessing. However, increasing the types and amount of external knowledge presented in the prompt about

---

\* Equal contribution
† Work done at University of Cambridge.
Code: https://github.com/cambridgeltl/POSQA

objects has a significant impact on the behaviour of LLMs. In particular, LLMs tend to develop their mental representation of objects referred to based on the given context in prompts rather than relying on their internal weights, even if the given context information is incorrect.

To conclude, our contributions can be summarized into three folds:

- We propose a simple but effective size comparison dataset to probe the real-world understanding of LLMs.

- We analyse the limits of the real-world understanding of LLMs with comprehensive prompt-based probing experiments.

- We discuss the vulnerability and the alignment of the world knowledge of LLMs.

## 2 Background

### 2.1 World Models and World Knowledge

There has been a wide and long-lasting debate about whether LLMs really have their internal world models and to what extent their world knowledge aligns with humans. Mind's Eye (Liu et al., 2022b) proposed to augment language models with an external physical simulation engine for better understanding the physical phenomena. RAP (Hao et al., 2023) suggests LLMs as both a world model and a reasoning agent and includes a principled planning algorithm for strategic exploration in a vast reasoning space. Xiang et al. (2023) deploys an embodied agent in a world model to endow LLMs with a diverse set of embodied experiences by fine-tuning. Although the world model of LLM can be effectively augmented and they indeed display a certain level of real-world understanding, there is still a lack of sufficient study on the boundary of the world understanding of LLMs. Similarly to other research on LLMs inspired by cognitive science and psychology (Binz and Schulz, 2022; Bisk et al., 2020a; Mahowald et al., 2023; Prystawski et al., 2022), we propose to audit the real-world understanding of LLM with questions as simple as size comparison.

### 2.2 Physical World Understanding Datasets

PIQA (Bisk et al., 2020b) is a popular data set for physical commonsense reasoning to benchmark progress in physical commonsense understanding.

PIQA dataset consists of more than 16,000 training QA pairs, with additional 2K and 3K held for development and testing. The task is multiple choice question answering: Given a question and two possible solutions, a model or a human must choose the most appropriate solution, of which exactly one is correct. MMLU-Physics (Hendrycks et al., 2021) contains 206 samples of physics consisting of multiple choice questions at the college and high school level to evaluate the academic and professional understanding of the model in the physics domain. UTOPIA (Liu et al., 2022b) is a new multi-task physics alignment dataset that aims to benchmark how well current LMs can understand and areas over some basic laws of physics. It leverages a physics engine to generate data for 39 subtasks covering six common scenes that involve understanding basic principles of physics. PROST (Aroca-Ouellette et al., 2021) is a new probing dataset to evaluate the ability of pre-trained LMs to understand and reason about the physical world. It contains 18,736 multiple-choice questions made from 14 manually curated templates, covering 10 physical reasoning concepts. The existing datasets contain questions from different dimensions, but they fail to effectively evaluate some particular aspect of the understanding of the physical world of LLMs. Since we want to probe the effect of context for in-context learning, it is necessary to have a content-controllable and dimension-specific dataset. Based on the requirements, we propose **POSQA**: a **P**hysical **O**bject **S**ize **Q**uestion-**A**nswering dataset which is also designed to test the size understanding ability of LLMs on physical world objects.

## 3 POSQA

POSQA consists of 12,000 multiple choice questions designed to probe the physical world understanding ability of the language model in the size dimension. We design two types of questions, each of them containing 6,000 questions. Table 1 shows the statistics of POSQA. The size comparison covers 92 entities, ranging from proton to universe. The entity and size information are obtained from Nikon Universcale, which aims to allow people to see and understand the relative size of the full range of known objects in our universe. We design four manually written templates to construct the two types of size questions. We show the templates

---

https://www.nikon.com/about/sp/universcale/

| Question Type | Bigger | Smaller | Total |
|---|---|---|---|
| General Question | 3,000 | 3,000 | 6,000 |
| Special Question | 3,000 | 3,000 | 6,000 |

Table 1: The statistics of POSQA.

in detail below.

**General Question** A general question requires the answer "yes" or "no". We use two templates to generate general questions.

TEMPLATE 1: Is *Entity A* bigger than *Entity B*?

TEMPLATE 2: Is *Entity A* smaller than *Entity B*?

We replace *Entity A* and *Entity B* with different entity names. For each template, we use the same Entity A – Entity B pair to generate a question, which is to avoid introducing bias. We generate 3,000 questions for each template, so there are 6,000 general questions in total. Based on the actual size of each entity, we label each question with "yes" or "no". The general questions aim to evaluate the size knowledge of objects contained in the LMs and the understanding of LMs on yes/no labels.

**Special Question** A special question begins with an interrogative word "which". We design two templates for special questions.

TEMPLATE 3: Which one is bigger between *Entity A* and *Entity B*?

TEMPLATE 4: Which one is smaller between *Entity A* and *Entity B*?

Similarly, we also use the same *Entity A – Entity B* pair to generate a question on each template. We generated a total of 6,000 questions, 3,000 questions for each template. The label of a special question is different from the general question. In the special question, we label each question with the actual entity name, which is exactly the same as *Entity A* or *Entity B*. The special questions are intended to test the understanding of the size of the LMs and the understanding of the LMs about the interrogative word 'which' of the question.

**Entity Feature** We collect the features of the 92 entities, including scale, size, magnitude, and text. The scale feature stores the size information of the entity in a specific size unit. For example, *the scale of the Solar System is 9 billion km and the scale of an Atom is 100 pm*. The size feature stores the absolute value of the size of the entity representing in scientific notation. The magnitude

feature represents the exponent of size which is stored in the size feature. The text feature contains a textual description of the entity.

## 4 Methodology

In this section, we cover the details of the proposed approach, first describing the designed prompt in Section 4.1, followed by the models used in the experiments in Section 4.2 and the introduction of the evaluation methods in Section 4.3.

### 4.1 Prompt Design

We construct different prompts that are used in our experiments.

#### 4.1.1 Plain Question Prompt

Plain question prompt is to query the model with a single plain question without any hint or knowledge. We aim to test how the model performs on POSQA without any external auxiliary information. The model answers the query purely based on the knowledge stored within its weights.

#### 4.1.2 Relevant Knowledge-augmented Prompt

External knowledge has been shown to be helpful for various NLP tasks, including common sense reasoning (Liu et al., 2022a). We consider two kinds of knowledge in our experiments: (1) Exact Size Information from POSQA (2) Generated Knowledge from GPT-3. Knowledge is considered as the context that is concatenated with a question. We use the knowledge-augmented prompt to query LLMs to see how the context would affect the model's prediction.

**Exact Size Information** The exact size information of each object in POSQA is stored as an entity feature. For a size comparison question, first we retrieve the exact size the two entities respectively, then we rewrite the original prompt to a knowledge-augmented prompt. In particular, we add a sentence to describe the size of the two objects before the question. The sentence is: The size of *Entity A* is *Exact Size of Entity A*. The size of *Entity B* is *Exact Size of Entity B*.

**Generated Knowledge** We generate the entity-related knowledge statement by querying an LLM. We consider two types of knowledge of entities: (1) general knowledge, which describes the general information of an entity; (2) size knowledge, which describes the size information of an entity. The purpose is to investigate which knowledge is more

useful to the model when answering the size-related questions. We query GPT-3 using the prompt 'Generate knowledge about *entity* in one sentence.' for extraction of general knowledge and 'Generate size knowledge of *entity* in one sentence.' for size knowledge extraction. The knowledge generated from GPT-3 is stored and used as a context in the knowledge-augmented prompt. Then we concatenate the knowledge generated with the size comparison question to query LLMs.

### 4.1.3 Adversarial Prompt with Knowledge Perturbation

In addition to the above useful prompt, we also design some adversarial prompts with knowledge perturbation. We aim to test how the model behaves when the given context is not useful or against the knowledge stored in its internal weights. We consider three settings: (1) Partial Information Provided (2) Masking Particular Information (3) Counterfactual Size Information.

**Partial Information Provided** In Exact Size Information Prompt, we provide the exact size information of two entities as the context. To investigate to what extent the model would utilise the context information, instead of giving two entity size information, we only provide one of them. This is to test whether the model could extract useful information from its internal weights to use together with the context information.

**Masking Particular Information** To further investigate whether the exact size helps the model when answering size comparison questions, we manually mask important information in the context. For example, in Exact Size Information Prompt, we mask the exact size or entities, respectively, to see the performance gap between using the masked prompt and the unmasked prompt. We replace the exact size of the entities with the mask token *[MASK]*.

**Counterfactual Size Information** Instead of providing the true size information in the context, we replace it with the wrong size information to investigate what predictions the model would make when the external context knowledge contradicts the knowledge stored in its weights. If the model will fully utilise the counterfactual size information when answering the size comparison questions. In particular, we swap the size information of the two entities in the prompt to mislead the model.

## 4.2 Models

Previous work (Wei et al., 2022a) (Sanh et al., 2022) has shown that instruction-tuned language models on a collection of NLP tasks formatted with instructions substantially improve the ability of language models to perform an unseen task from an instruction, especially zero-shot performance. In this work, we do experiments on three kinds of instruction-tuned model: Flan-T5 (from 80M to 3B) (Chung et al., 2022), InstructGPT (175B) (Ouyang et al., 2022), and recent ChatGPT. Flan-T5 is instruction-tuned on 1,836 NLP tasks that initialise from prior public checkpoints of T5 (Raffel et al., 2020). InstructGPT uses reinforcement learning from human feedback (Christiano et al., 2017) (Stiennon et al., 2020) to fine-tune GPT-3 (Brown et al., 2020) to follow a broad class of written instructions. ChatGPT uses the same training methods as InstructGPT, but with slight differences in data collection setup. It can interact in the form of a conversational dialogue and provide human-like responses.

## 4.3 Evaluation

In this part, we describe the evaluation process and the evaluation metrics we use.

### 4.3.1 Answer Mapping

Since we query the LLMs to generate the answer to the question, it cannot be guaranteed that all the generated answers are exactly the labels. We use an answer mapping process to map the generated answer to the answer label. For general questions, the labels are yes or no. If the predicted answer contains 'yes'/'YES', we assume its predicted label is yes. If the predicted answer contains the 'no'/'NO', we assume that its predicted label is no. For special questions, the labels are entity names. We calculate the Levenshtein distance (Li and Liu, 2007) between the predicted entity and the two candidate entities, respectively. The Levenshtein distance is a string metric for measuring the difference between two sequences, and a smaller distance means the two strings are more similar. We choose the candidate entity with smaller Levenshtein distance as the predicted label.

### 4.3.2 Metrics

We consider four metrics in our experiments. **Accuracy** and **Macro-F1** scores are two commonly

---

https://openai.com/blog/chatgpt/

| Prompt | FT5-Small | | FT5-Base | | FT5-Large | | FT5-XL | | GPT3-Davinci | | GPT3.5-Turbo | |
|---|---|---|---|---|---|---|---|---|---|---|---|---|
| Metric | Acc | Macro-F1 | Acc | Macro-F1 | Acc | Macro-F1 | Acc | Macro-F1 | Acc | Macro-F1 | Acc | Macro-F1 |
| Plain Question | 0.51 | 0.34 | 0.50 | 0.35 | 0.56 | 0.54 | 0.58 | 0.51 | 0.52 | 0.38 | 0.69 | 0.69 |
| + General Knowledge Information | 0.50 | 0.34 | 0.52 | 0.52 | 0.64 | 0.63 | 0.67 | 0.67 | 0.51 | 0.35 | 0.79 | 0.79 |
| + Size Knowledge Information | 0.50 | 0.40 | 0.55 | 0.53 | 0.63 | 0.61 | 0.76 | 0.76 | 0.56 | 0.48 | 0.85 | 0.85 |
| + Exact Size Information | 0.50 | 0.50 | 0.50 | 0.54 | 0.62 | 0.77 | 0.87 | 0.88 | 0.73 | 0.89 | 0.90 | 0.89 |
| + Only Head Entity Gold Size | 0.50 | 0.34 | 0.54 | 0.50 | 0.63 | 0.60 | 0.64 | 0.64 | 0.51 | 0.35 | 0.69 | 0.69 |
| + Only Tail Entity Gold Size | 0.50 | 0.34 | 0.49 | 0.34 | 0.55 | 0.54 | 0.66 | 0.63 | 0.51 | 0.34 | 0.75 | 0.74 |
| + Masking Size Information | 0.50 | 0.49 | 0.49 | 0.34 | 0.53 | 0.41 | 0.74 | 0.74 | 0.53 | 0.39 | 0.80 | 0.80 |
| + Masking Entity Information | 0.50 | 0.33 | 0.50 | 0.37 | 0.49 | 0.42 | 0.63 | 0.58 | 0.52 | 0.46 | 0.70 | 0.69 |
| + Counterfactual Size Information | 0.50 | 0.47 | 0.49 | 0.33 | 0.47 | 0.33 | 0.23 | 0.23 | 0.42 | 0.41 | 0.38 | 0.38 |

Table 2: The results of using different prompt settings on various models on the general question of POSQA.

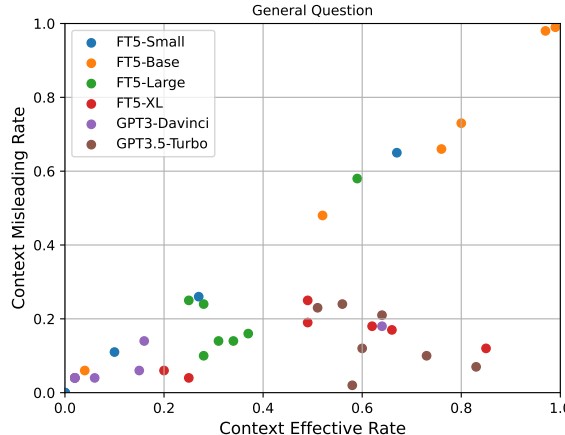

Figure 1: The results of CER and CMR using different knowledge-augmented prompts (except for Counterfactual Size Information) on various models on general question of POSQA.

used metrics for the evaluation of the performance of the model in classification tasks. Accuracy considers global precision and recall of the categories, while Macro-F1 computes the average of the F1 scores obtained by individual categories.

To explore the influence of context on the prompt, we propose two quantitative evaluation metrics: **Context Effective Rate (CER)** and **Context Misleading Rate (CMR)**. We calculate CER and CMR by comparing the output of a model using a prompt that contains context information with the output of a model using a prompt that does not contain any context information. Specifically, CER evaluates how many incorrectly answered questions can be correctly answered after adding the context in the prompt. CMR evaluates how many correctly answered questions can be incorrectly answered after adding the context to the prompt.

## 5 Results and Analysis

### 5.1 Baseline

The baseline is to query LLMs using the Plain Question prompt and the results are presented in Table 2 for general questions and Table 3 for special questions. When the number of parameters of the model exceeds 250M (Flan-T5-Base), the ability to answer the special size comparison questions begins to emerge. GPT3.5-Turbo achieves the best accuracy score and Macro-F1 score on both types of questions. It is surprising that significantly smaller models, such as Flan-T5-Large (780M) and Flan-T5-XL (3B), exhibit superior performance. For example, Flan-T5-XL outperforms GPT3-Davinci (175B) by 0.06 precision in answering general questions and 0.03 precision in answering special questions. In general, LLMs perform better at special questions. For example, the accuracy increases by 0.14 on Flan-T5-XL and 0.04 on GPT3.5-Turbo, respectively, from answering general questions to special questions.

Our empirical investigation also reveals that the GPT3-Davinci model tends to provide an initial incorrect answer, despite subsequently offering a correct explanation for the given question. This phenomenon occurs especially when directly querying GPT3-Davinci with general questions to get the "yes" or "no" answers. We speculate that it could be attributed to the use of first-word sampling techniques during the decoding process. However, this phenomenon does not occur in GPT3.5-Turbo which has been optimised for dialogue scenario. Even GPT3.5-Turbo, one of the most powerful LLMs, can only achieve an average 0.71 accuracy score on these two types of questions. This suggests that although larger LMs may possess certain advantages in most situations, they may still lack real-world understanding when it comes to answering basic size comparison questions.

| Models→ | FT5-Small | | FT5-Base | | FT5-Large | | FT5-XL | | GPT3-Davinci | | GPT3.5-Turbo | |
|---|---|---|---|---|---|---|---|---|---|---|---|---|
| Prompt↓, Metric→ | Acc | Macro-F1 | Acc | Macro-F1 | Acc | Macro-F1 | Acc | Macro-F1 | Acc | Macro-F1 | Acc | Macro-F1 |
| Plain Question | 0.51 | 0.49 | 0.51 | 0.50 | 0.63 | 0.62 | 0.72 | 0.71 | 0.69 | 0.69 | 0.73 | 0.72 |
| + Size Knowledge Information | 0.52 | 0.50 | 0.55 | 0.53 | 0.74 | 0.73 | 0.79 | 0.78 | 0.76 | 0.76 | 0.83 | 0.83 |
| + General Knowledge Information | 0.52 | 0.50 | 0.54 | 0.52 | 0.67 | 0.66 | 0.72 | 0.71 | 0.71 | 0.71 | 0.80 | 0.80 |
| + Exact Size Information | 0.52 | 0.50 | 0.56 | 0.54 | 0.77 | 0.77 | 0.88 | 0.88 | 0.89 | 0.89 | 0.90 | 0.89 |
| + Only Head Entity Gold Size | 0.50 | 0.48 | 0.52 | 0.51 | 0.68 | 0.67 | 0.77 | 0.77 | 0.58 | 0.57 | 0.74 | 0.73 |
| + Only Tail Entity Gold Size | 0.51 | 0.49 | 0.52 | 0.51 | 0.64 | 0.63 | 0.77 | 0.76 | 0.69 | 0.68 | 0.76 | 0.76 |
| + Masking Size Information | 0.52 | 0.49 | 0.51 | 0.49 | 0.68 | 0.68 | 0.76 | 0.76 | 0.67 | 0.67 | 0.64 | 0.63 |
| + Masking Entity Information | 0.51 | 0.47 | 0.54 | 0.52 | 0.71 | 0.69 | 0.84 | 0.84 | 0.84 | 0.83 | 0.82 | 0.82 |
| + Counterfactual Size Information | 0.52 | 0.50 | 0.49 | 0.47 | 0.32 | 0.32 | 0.18 | 0.18 | 0.25 | 0.25 | 0.29 | 0.29 |

Table 3: The results of using different prompt settings on various models on the special question of POSQA.

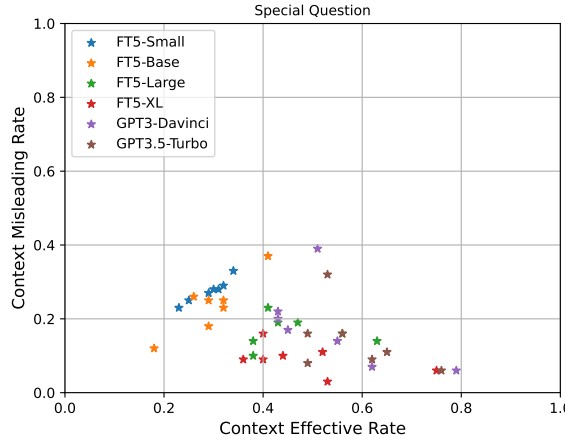

Figure 2: The results of CER and CMR using different knowledge-augmented prompts (except for Counterfactual Size Information) on various models on special question of POSQA.

## 5.2 Prompts with Augmented Knowledge

The performance of using Relevant Knowledge-Enhanced Prompt as contextual information to query LLMs is shown in Table 2 and Table 3. According to the presented results, augmenting the original size comparison questions with supplementary knowledge about the mentioned objects can significantly enhance an LLM's performance, suggesting that these models can effectively use contextual information to improve their real-world understanding. In particular, as the model size scales beyond 780M (Flan-T5-Large), its ability to utilise prompt information increases greatly. Among these three Knowledge-augmented prompts, adding exact size information of the two compared objects in the context is the most effective way, which adheres to the intuition. By using this prompt, GPT3.5-Turbo achieves 0.9 accuracy score on both general questions and special questions, which shows great ability in utilising the exact size information in the context. GPT3-Davinci also obtains the 0.89 ac-

curacy score and the Macro-F1 score on special questions. Even for FT5-XL, it can achieve 0.87 accuracy score and 0.88 Macro-F1 score on general questions and special questions, respectively, which is quite close to GPT3.5-Turbo with regard to the size comparison ability.

In addition to Gold Size Information, the generated knowledge information and size information are also helpful to LLMs in answering the size comparison question. The results also show that size knowledge is more effective than general knowledge as contextual information. Specifically, on GPT3.5-Turbo, using size knowledge information can increase the accuracy and Macro-F1 scores by 0.06 on general questions and 0.03 on special questions from using general knowledge information.

Although there is a significant improvement after providing LLMs with useful contextual information, the result still fall short of the human-level understanding of the real world, especially when adding the ground truth exact size information. Furthermore, the LLMs' performance in answering different types of question has an obvious difference, despite the two types of questions being semantically equivalent for humans. This observation indicates that LLMs are more sensitive to question formats than humans. In summary, the findings suggest that supplementing original questions with additional information can enhance an LLM's real-world understanding. However, even with this augmentation, LLMs' ability to achieve human-level understanding of the real world is still limited, and their sensitivity to question formats remains a challenge.

## 5.3 Context vs. Weights

By using the Adversarial Prompt with Knowledge Perturbation, we further explore the influence of the contextual information. The results in Table 2 and Table 3 provide valuable insights into how providing additional information can help improve LLM's

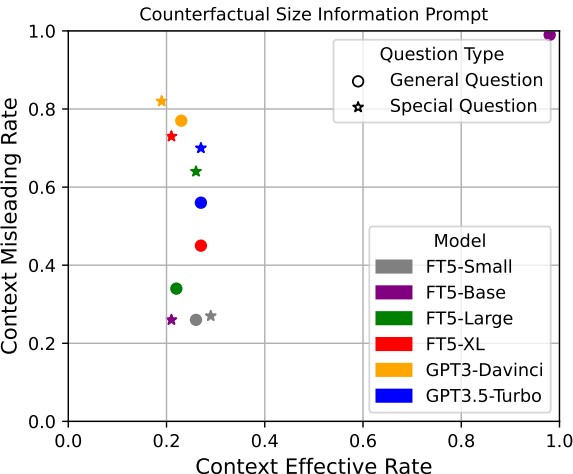

Figure 3: The results of CER and CMR using Counterfactual Size Information prompt on various models on two types of questions of POSQA.

real-world understanding. When we only provide LLMs with partial information, the performance drops significantly compared to jointly providing information about both objects. For instance, on GPT3.5-Turbo, the accuracy score decreases by 0.21 on general questions and by 0.16 on special questions when only providing the exact size information of the head entity. The results also reveal an imbalance in the utilisation of information, as the LMs tend to benefit more from extra information about the tail objects than the head objects.

Masking either of the size or entity information would influence the performance of the LLMs. Interestingly, masking size information decreases the performance more in answering the special questions, while masking entity information decreases the performance more in answering the general questions. For example, on GPT3.5-Turbo, the accuracy score is 0.8 on general questions while only 0.64 on special questions when masking size information. When masking entity information, GPT3.5-Turbo only gets a 0.7 accuracy score on general questions while 0.82 on special questions. Even when key information is masked, the context still provides some useful information which LLMs can utilise when answering on one type of question.

Figure 1 and Figure 2 show the CER and CMR results using different knowledge-augmented prompts (except for Counterfactual Size Information) on various models on general and special questions, respectively. As the scale of LMs grows, the

ability of LMs to utilise contextual information in the prompt is also enhanced. For example, the results of GPT3.5-Turbo are scattered in the lower right of the diagram, which means that a powerful LLM should have high CER and low CMR.

When providing LLMs with counterfactual size information that is not helpful to answering size comparison questions, accuracy and Macro-F1 scores decrease to a large extent. It should be noted that FT5-XL has the lowest accuracy score and Macro-F1 score on both types of questions, with only 0.23 and 0.18 respectively. Figure 3 illustrates the results of CER and CMR using the Counterfactual Size Information prompt on various models for two types of questions. The CER is low on all models, while the CMR increases as the scale of the models grows. It shows a great ability to utilise counterfactual contextual information for larger LMs (e.g. GPT3-Davinci), which is revealed by the low CER and high CMR on general and special questions.

The results also highlight the importance of contextual information in LLMs' real-world understanding, as providing counterfactual information significantly decreases their performance with high CMR in answering both general and special questions. This indicates that the LLMs rely more on contextual information rather than their internal weights learnt during pre-training, which reflects their in-context learning capabilities. In summary, the findings provide valuable insights into the strengths and limitations of LLMs in real-world understanding. They demonstrate that while providing additional information can enhance LLMs' performance and their sensitivity to contextual information. However, this also raises concerns about the robustness of LLMs' real-world understanding, as they may be easily induced to perform harmful actions in real-world scenarios.

## 6  Discussion

**LLMs' ability to understand the size of physical objects in the real world remains a challenge.** Our experiments underscore that even the most advanced LLMs at our disposal struggle to consistently grasp the sizes of physical objects. Specifically, GPT3.5-Turbo registers an average accuracy score of 0.71 when directly addressing the two question types present in POSQA. This performance reveals a pronounced disparity compared to human comprehension of size, particularly when

object size information is explicitly provided. Humans, when confronted with size comparison tasks, often engage in mental simulations, drawing upon their accumulated knowledge to envisage the sizes of objects (de Koning et al., 2017b). For a human, the ability to tackle size comparison effectively hinges on possessing adequate size-related information about the objects in question. Similarly, LLMs should leverage the knowledge encoded in their internal weights to adeptly respond to size comparison queries.

**LLMs prefer to utilise the information in the given context rather than knowledge stored in their internal weights.** Our experiments demonstrate that giving useful information in a prompt can enhance the performance of LLMs. For instance, GPT3.5-Turbo achieves 0.9 accuracy score on both two types of questions. However, LLMs cannot make good use of the external context information when they are only given partial information. It is noteworthy that adding error information to the prompt largely decreases the performance of LLMs. For example, GPT3.5-Turbo can only get 0.38 Accuracy and Macro-F1 score on general questions and 0.29 Accuracy and Macro-F1 score on special questions. Even research (Shi et al., 2023) has shown that adding an instruction to ignore irrelevant information brings performance gains; A single piece of irrelevant information can distract the models and substantially degrade their performance. These results indicate that the context in the prompt is extremely important for LLMs and that LLMs will utilise the information in the context.

**LLMs are sensitive to the format of the query, even if they are semantically equivalent.** In our experiments, we query LLMs with different format of size comparison questions from POSQA. The results show that LLMs are not robust enough when faced with the same semantics, but different forms of the queries. For example, on FT5-XL, it achieves a 0.58 accuracy score on general questions versus a 0.72 accuracy score on special questions. Based on the behaviour of the LLMs on answering size comparison questions, it is not certain which forms of questions the models are better at solving, and the performance is also influenced by the context added to the prompt. Although the performance gap between these two types of questions in GPT3.5-Turbo has narrowed, it is still a noteworthy problem when training robust LLMs in the future.

| Krippendorff's Alpha | Human (Online) | Human (Offline) |
|---|---|---|
| Human (Online) | 0.740 | 0.780 |
| Human (Offline) | 0.780 | 0.791 |
| ChatGPT | 0.644 | 0.687 |

Table 4: The internal and mutual consistency among human annotators in different settings and ChatGPT is measured by Krippendorff's Alpha.

**Alignment of World Models** To further investigate the alignment of LLM world models, we randomly sampled 100 examples and annotated them with four human annotators in two settings: (1) Online: Annotators are free to access any external or online sources of knowledge, and (2) Offline: Annotators are prohibited from relying on external resources during annotation. Surprisingly, online annotations lag behind offline annotations in terms of accuracy, with online accuracy at 0.86 and offline accuracy at 0.88. ChatGPT is slightly behind human performance, reaching an accuracy of 0.77. As shown in Table 4, we compute the Krippendorff Alpha (Krippendorff, 2011) to assess the internal and mutual agreement between the online and offline annotators and ChatGPT. The scores indicate that ChatGPT is more consistent with human annotators without accessing an external knowledge source, which arouses curiosity that ChatGPT may share characteristics and bias similar to that of human intuition or fast thinking, which could be further investigated from the psycholinguistic point of view.

## 7 Conclusion

We propose POSQA: a Physical Object Size Question-Answering dataset with two types of size comparison questions to probe the ability of LLMs to understand the size of physical world objects. We design different knowledge-augmented prompt settings to investigate the effect of the context in the prompt. Our experiments demonstrate that LLMs still fail to demonstrate a robust understanding of the size of physical objects. The ability of LLMs to understand the size of physical objects in the real world remains a challenge for the future. The results also show that LLMs prefer to utilise the information in the given context rather than to use the knowledge stored in their internal weights. This also raises concerns about the robustness of LLMs' understanding ability to identify the useful and correct contextual information in the prompt.

## Acknowledgement

We are grateful to acknowledge that the work of the joint first author, CS, has been jointly supported by a donation from Toshiba Europe and the Engineering and Physical Sciences Research Council of UKRI (grant number 2752931).

## Limitations

Our datasets comprise a modest total of 92 objects. Consequently, rather than serving as comprehensive evaluation toolkits that encapsulate the breadth of LLM world models, they may be best suited for probing or auditing LLM performance in real-world understanding, preferably in tandem with broader benchmarks. Additionally, the dataset only captures rudimentary relationships between two objects—specifically, size comparisons. The incorporation of more intricate interactions and dynamics among multiple objects might provide a deeper insight into the LLM world model. Moreover, due to resource constraints, our experiments were limited to Flan-T5, GPT3-Davinci, and GPT3.5-Turbo.

## Ethics Statement

The purpose of this research project is to evaluate the world understanding capabilities of Language Models (LLMs) by synthesizing new datasets from existing knowledge bases, with the aim of advancing Natural Language Processing (NLP) research and improving LLM performance. We are committed to conducting this research with the highest ethical standards, ensuring privacy and ethical considerations. No personally identifiable information or sensitive data is collected, stored, or processed, as the datasets are solely derived from publicly available knowledge bases and are anonymized. We actively mitigate biases in the underlying data by carefully selecting and preprocessing the knowledge base. The datasets created will be used exclusively for evaluating LLM world understanding and will not be used for any commercial, discriminatory, or unethical purposes. We prioritize responsible data usage, securely storing the data and making it accessible only to authorized researchers. Transparency and reproducibility are key, as we document the dataset synthesis process for others to reproduce and validate the results. Our research adheres to ethical guidelines, institutional policies, industry standards, and relevant regulations. Additionally, we foster collaboration and knowledge sharing within the research community, seeking to develop LLMs that better engage with the world and benefit society. Continuous evaluation and improvement are integral to our approach, and we welcome feedback from the research community and the wider public, as it contributes to the responsible development and application of LLMs, aligning with principles of fairness, privacy protection, transparency, and public benefit.

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

# A Appendix

## A.1 Error Analysis

Empirically, we realised that the error counts vary largely regarding to different entity mentions in the prompts. According to Figure 4, the AIDS virus is the most challenging entity for LLMs to distinguish the physical size relationship with other objects, compared to that of Mars, which is the easiest. Furthermore, as shown in Figure 5, it is more difficult for LLMs to correctly compare two objects with similar size magnitude.

## A.2 GPT3-Davinci using Chain-of-Thought

Existing research (Kojima et al., 2022)(Wei et al., 2022b) has shown that the use of CoT could significantly improve the performance of LLMs. We randomly sample 100 test cases (50 questions for each type) and report the results using Zero-Shot-CoT (Kojima et al., 2022) on GPT3-Davinci in Table 5. The results show that after using CoT, accuracy and Macro-F1 scores are notably improved across all settings on both types of questions.

## A.3 GPT3-Davinci using Different Temperature Parameters

To investigate the effect of different temperature parameters, we performed experiments at three different temperatures (0, 0.5, 1) in 100 randomly sampled test cases (50 questions for each type). This

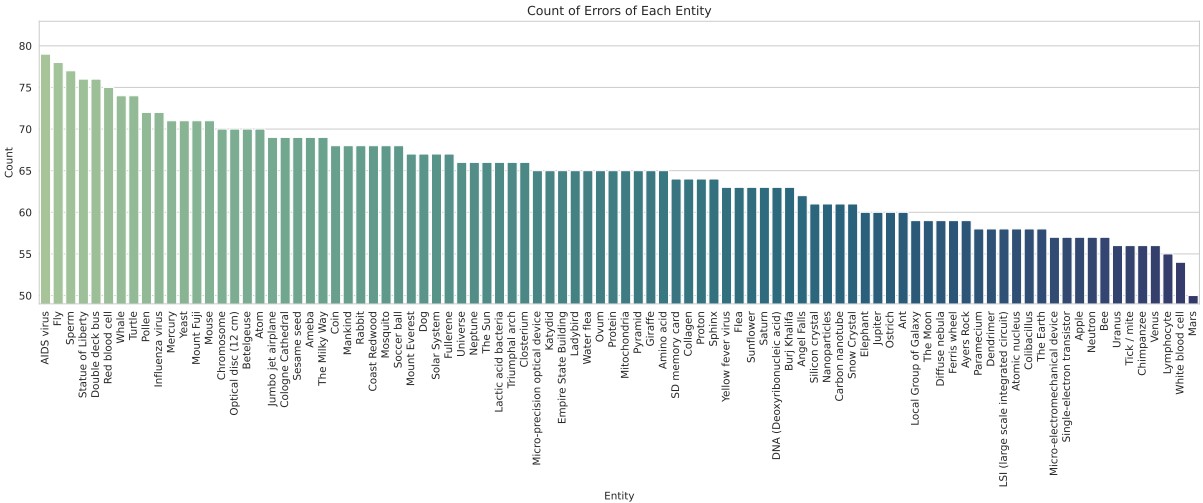

Figure 4: The number of mistakes related to each entity.

| Question | Prompt | Accuracy | Macro-F1 | Accuracy | Macro-F1 |
|---|---|---|---|---|---|
| | | w/o CoT | | with CoT | |
| General | Plain Question | 0.52 | 0.38 | 0.62 | 0.62 |
| | + Gold Size Information | 0.78 | 0.78 | 0.88 | 0.88 |
| | + General Knowledge Information | 0.50 | 0.33 | 0.66 | 0.65 |
| | + Size Knowledge Information | 0.62 | 0.56 | 0.76 | 0.76 |
| Special | Plain Question | 0.84 | 0.83 | 0.90 | 0.87 |
| | + Gold Size Information | 0.94 | 0.93 | 0.94 | 0.93 |
| | + General Knowledge Information | 0.84 | 0.81 | 0.92 | 0.91 |
| | + Size Knowledge Information | 0.88 | 0.86 | 0.94 | 0.93 |

Table 5: The results of GPT3-Davinci using Chain-of-Thought (CoT) in different prompts on 100 randomly chosen test cases (50 questions for each type).

result is shown in Table 6. The results show that setting the temperature to 0 can give the best results. To alleviate the randomness effect of temperature, in all our experiments, we set the temperature as 0, so that the LLMs become deterministic.

## A.4 GPT3-Davinci using Gold Size Information Prompt Templates

Since LLMs is sensitive to the format of the prompt, we use different prompt templates to probe their influence. We mainly tested the gold size information prompt template and we list all templates in Table 7. We conducted experiments on the same 100 randomly sampled test cases (50 questions for each type), and the result is shown in Table 8. The effect of using different templates is notable in the general question, and using Template 3 has the best performance. Interestingly, Template 4 decreases the performance on both general and special questions, which means that the order of the question and context is significant when choosing the prompt.

## A.5 Statistical Significance

We assess the statistical significance of the differences or relationships observed in our experiments using a two-sample t-test (Cressie and Whitford, 1986). The results in Table show that the p-value ($p < 0.05$) is sufficiently low to reject the null hypothesis, indicating that there is a significant difference between the various experiment settings with GPT3.5-Turbo on the POSQA data set.

| Question | Prompt | Accuracy | Macro-F1 | Accuracy | Macro-F1 | Accuracy | Macro-F1 |
|---|---|---|---|---|---|---|---|
| Temperature | | | 0 | | 0.5 | | 1 |
| General | Plain Question | 0.52 | 0.38 | 0.52 | 0.38 | 0.52 | 0.38 |
| | + Gold Size | 0.78 | 0.78 | 0.80 | 0.80 | 0.78 | 0.78 |
| | + General Knowledge | 0.50 | 0.33 | 0.50 | 0.33 | 0.50 | 0.33 |
| | + Size Knowledge | 0.62 | 0.56 | 0.62 | 0.56 | 0.60 | 0.52 |
| Special | Plain Question | 0.84 | 0.83 | 0.82 | 0.80 | 0.82 | 0.81 |
| | + Gold Size | 0.94 | 0.93 | 0.90 | 0.90 | 0.92 | 0.92 |
| | + General Knowledge | 0.84 | 0.81 | 0.86 | 0.83 | 0.82 | 0.78 |
| | + Size Knowledge | 0.88 | 0.86 | 0.82 | 0.78 | 0.80 | 0.75 |

Table 6: The results of GPT3-Davinci using different temperatures in different prompts on 100 randomly chosen test cases (50 questions for each type).

| | |
|---|---|
| Template 1: | The size of *Entity A* is *Exact Size of Entity A*. The size of *Entity B* is *Exact Size of Entity B*. *+ Question* |
| Template 2: | *Entity A*: *Exact Size of Entity A*; *Entity B*: *Exact Size of Entity B*. *+ Question*. |
| Template 3: | *Entity A* is *Exact Size of Entity A* and *Entity B* is *Exact Size of Entity B*. *+ Question* |
| Template 4: | *Question* + The size of *Entity A* is *Exact Size of Entity A*. The size of *Entity B* is *Exact Size of Entity B*. |

Table 7: Different gold size information prompt templates used in Table 8.

| Question | Prompt | Accuracy | Macro-F1 |
|---|---|---|---|
| General | Plain Question | 0.52 | 0.38 |
| | + Gold Size Information using Template 1 | 0.78 | 0.78 |
| | + Gold Size Information using Template 2 | 0.82 | 0.82 |
| | + Gold Size Information using Template 3 | 0.88 | 0.88 |
| | + Gold Size Information using Template 4 | 0.50 | 0.37 |
| Special | Plain Question | 0.84 | 0.83 |
| | + Gold Size Information using Template 1 | 0.94 | 0.93 |
| | + Gold Size Information using Template 2 | 0.94 | 0.95 |
| | + Gold Size Information using Template 3 | 0.90 | 0.89 |
| | + Gold Size Information using Template 4 | 0.86 | 0.84 |

Table 8: The results of GPT3-Davinci using different gold size information prompt templates on 100 randomly chosen test cases (50 questions for each type).

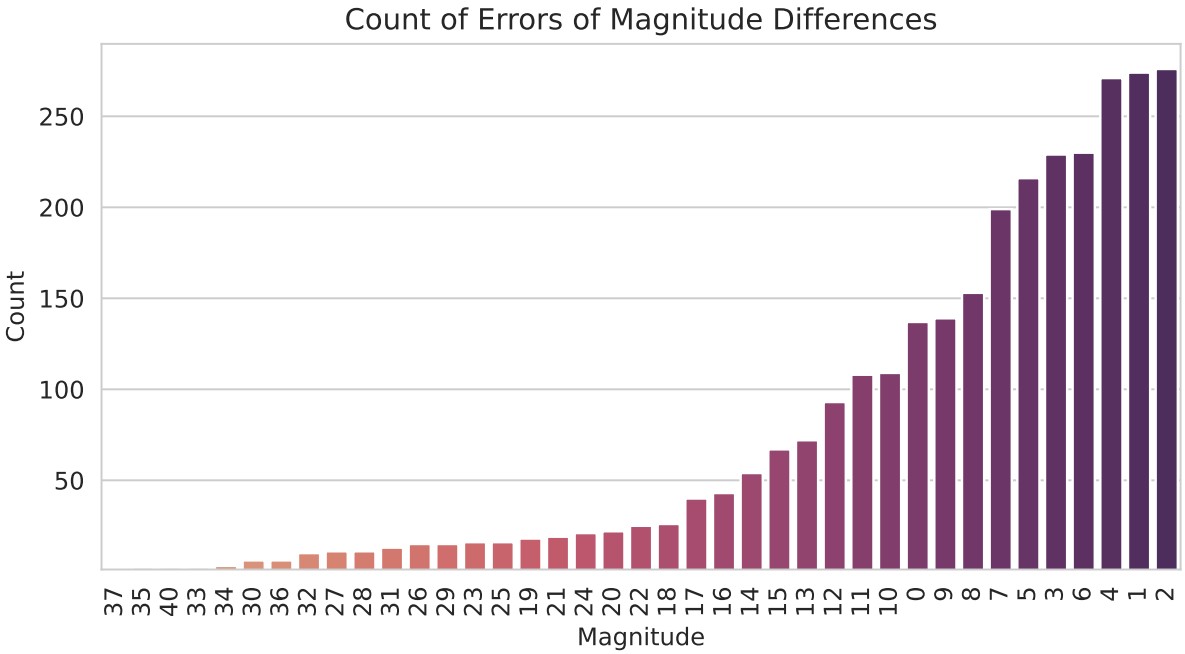

Figure 5: The number of mistakes related to the magnitude difference between two entities.