# OpenReview forum: "POSQA: Probe the World Models of LLMs with Size Comparisons"
_EMNLP/2023/Conference — EMNLP 2023 Findings_

### Official Review · Reviewer_266w · 2023-07-30

**Soundness:** 3

**Excitement:**

3: Ambivalent: It has merits (e.g., it reports state-of-the-art results, the idea is nice), but there are key weaknesses (e.g., it describes incremental work), and it can significantly benefit from another round of revision. However, I won't object to accepting it if my co-reviewers champion it.

**Missing References:**

Previous research has curated a similar dataset consisting of 486 pairs involving 41 objects. While this study presents a more extensive collection, it is essential to acknowledge and cite the preceding work in a related domain.

Hessam et al. "Are elephants bigger than butterflies? Reasoning about sizes of objects." In Proceedings of the Thirtieth AAAI Conference on Artificial Intelligence (AAAI'16). https://arxiv.org/abs/1602.00753

**Paper Topic And Main Contributions:**

This paper introduces a new dataset designed to assess the knowledge of Large Language Models (LLMs) concerning the physical sizes of different real-world objects. The authors gathered data on 92 entities and constructed an extensive set of pairwise questions that compare the sizes of these entities.

The study involves two types of questions: "General Questions," which require yes/no answers, and "Specific Questions," which demand identifying the entity that is bigger/smaller. To enhance the prompts, the authors incorporated in-context demonstrations using either manually curated size information or knowledge pieces generated by an LLM.

The authors also created variations of the dataset by introducing perturbations in the quality of the information provided in the demonstrations. These perturbations involve withholding information about certain entities, masking spans of information, or including counterfactual statements.

The experimental results revealed that even though powerful models like GPT3.5-Turbo achieved up to 90% accuracy when given the gold size information (and 94% with CoT prompts), their parametric knowledge falls short compared to human knowledge, despite their extensive pretraining on general domain knowledge.

**Questions For The Authors:**

Question A: The linked website for the data (Footnote 1, www.posqa.data) is unreachable. Is this on purpose / will it be available after acceptance?

Question B: In line 214, you describe the entity features as "scale, size[, magnitude, and text]". I find it difficult to truly distinguish between the first two, as you mention that "the scale of the Solar System is 9 billion km", which I would intuitively label as its size rather than simply scale. Could you perhaps add an example entry of the entities dataset that showcases all four features for that entity?

Question C: Which GPT3 model are you using to generate the knowledge information in Section 4.1.2 (lines 246/269/274)? And similarly, when referring to InstructGPT and ChatGPT (lines 326-328), it would be nice to mention the exact name of each model in the API for further reproducibility.

Question D: What does it mean for models to have (near) 1 CER & CMR in Figures 1 and 3? I find these data points abnormal without being addressed in the text.

Question E: In line 410 you mention the use of "first-word sampling" techniques during decoding. What is this referring to? I don't see how this differs between InstructGPT and ChatGPT.

Question F: In lines 622-626, the finding that annotators with access to online resources perform worse on the task is remarkably counter-intuitive. Do you have any explanation on why this may be the case?

**Reasons To Accept:**

The paper presents a novel dataset aimed at evaluating the parametric knowledge of Large Language Models on a specific type of questions that has not been extensively explored in previous research. The probing setup utilized in the study is well-structured and methodologically sound. Furthermore, the conclusions drawn from the experimental results are well-supported and credible.

**Reasons To Reject:**

The paper mentions the study of the alignment of LLM knowledge with human comprehension of the world as one of its primary contributions. However, this aspect is only briefly addressed in the discussion section, where a small-scale experiment is conducted. Unfortunately, the analysis of this experiment is limited, as it solely compares ChatGPT's performance with that of human annotators.

Furthermore, the paper contains several grammatical errors that may impede comprehension, particularly for non-native English speakers. To enhance the paper's clarity, I suggest addressing the listed grammatical errors in the Presentation Improvements field and conducting a more thorough analysis of the experiment comparing LLMs' knowledge to human understanding. This could provide deeper insights into the alignment or disparities between the two.

**Reproducibility:**

4: Could mostly reproduce the results, but there may be some variation because of sample variance or minor variations in their interpretation of the protocol or method.

**Reviewer Confidence:**

3: Pretty sure, but there's a chance I missed something. Although I have a good feel for this area in general, I did not carefully check the paper's details, e.g., the math, experimental design, or novelty.

**Typos Grammar Style And Presentation Improvements:**

I would strongly suggest the use of a grammar/spell checker for the camera-ready version if the paper gets accepted, as most these issues should be easy to spot/fix.

Line 67: \citep instead of \citet for "de Koning et al."

Line 89: contributes -> contributions

Line 101: debates -> debate

Line 104: are aligned -> is aligned

Line 168: question -> questions

Line 349: 'yes' or 'yes' -> 'yes' or 'YES'

Tables 2/3: The Macro-F1 column should be center aligned to match the rest.

Tables 2/3: "+ Gold Size Information" is originally introduced as "Exact Size Information" in the Methodology (line 251), so it might be preferable to be consistent for ease of understanding.

Tables 2/3: Since you often compare the metrics between the two tables, would it not make more sense to combine them into one table? Currently, it is tedious to compare the performance of General vs Special questions since you need to check both tables to see the difference between the two formats. You could, for example, present the results as XX.XX / YY.YY, with XX corresponding to General and YY to Special (and adding this information to the caption).

Figures 1/2: I feel like they are misplaced since they are only discussed on Page 7 and the reader has to scroll back and forth to see what you are referring to.

Figures 1/2: It would be nice to have different shapes to indicate which experiment each data point corresponds to since now we can only discern between models. I am curious to see if, for example, the CER/CMR are higher when providing masked information vs gold sizes.

Line 388: reaches -> surpasses (since the performance starts increasing from FlanT5-Large onwards, while FlanT5-Base shows scores that are rather close to those of FlanT5-Small)

Throughout Section 5: Mirco-F1 -> Accuracy (You don't list Micro-F1 anywhere in Section 4.3.2?)

Line 392: surprised -> surprising

Lines 396/397/401/456: 0.6 -> 0.06, 0.3 -> 0.03, 0.4 -> 0.04, 0.6 -> 0.06, 0.3 -> 0.03

Line 499: examples -> example

Line 505: Even **when** the key information is masked [...]

Line 509 Figure 5 -> Figure 2 (?)

Line 522: "Other LMs still have the same tendency" (which LMs is this referring to? No model was mentioned before this sentence.)

Lines 546-547: They demonstrate that ~~while~~ providing extra information [...]

Line 943: Figure 5 -> Figure 4 (?)

---

> ### Author Rebuttal · Authors · 2023-08-29
>
> We extend our heartfelt appreciation for the insightful assessment of our paper. We value your valuable feedback and insights, and would like to address your observations and queries while providing further clarification on our research:
>
> **Addressing the Concerns:**
>
> - **Alignment of LLM Knowledge with Human Comprehension:** Exploring a large-scale alignment study between humans and LLMs is indeed an intriguing avenue, contingent upon the availability of resources within an academic context. Our current work, as showcased here, contributes valuable insights into comprehending the limitations of LLMs, delving into diverse analytical facets. It serves as a valuable resource that can potentially pave the way in this particular direction.
>
> - **Grammatical Errors and Clarity:** We are grateful for your suggestions and will fix the mentioned presentation issues and proofread the final version.
>
> **Responding to Questions:**
>
> - **Question A:** We express our gratitude for bringing the data link issue to our attention. In response, we are pleased to provide access to the [preview version of POSQA](https://anonymous.4open.science/r/POSQA-EF54/datasets). Upon acceptance, we commit to releasing the dataset on the Hugging Face Datasets platform, thereby facilitating extensive usage and validation.
>
> - **Question B:** We value your inquiry about distinguishing between "scale" and "size" in entity features. Below is an illustrative example of the entity "Solar System." We will also incorporate this example within our paper:
>
> ```json
> {
>   "scale": "9 billion km",
>   "size": "9×10^12",
>   "magnitude": "12",
>   "name": "Solar System",
>   "text": "The Solar System consists of the central Sun, eight planets (Mercury, Venus, Earth, Mars, Jupiter, Saturn, Uranus and Neptune), dwarf planets including Pluto and smaller celestial bodies. The distance from the Sun to Neptune, the farthest planet from the Sun, is about 4.5 billion km. The Solar System was formed about 4.6 billion years ago when interstellar clouds, which were floating in space, were pulled together by gravity and began to rotate, pulling mass into the centre to form a primordial solar nebula. The other planets are believed to have formed from the gas and dust which remained in the surrounding area."
> }
> ```
>
> - **Question C:** Thank you for your suggestion. The InstructGPT(GPT3) we employed is text-davinci-003, and the ChatGPT variant we utilized is gpt-3.5-turbo. We will provide clarification on this matter within our paper.
>
> - **Question D:** In essence, CER evaluates how many incorrectly answered questions can be corrected after adding context to the prompt. CMR, on the other hand, assesses how many correctly answered questions can turn incorrect upon incorporating context. When CER & CMR values approach 1, it signifies that all accurately answered questions are susceptible to being incorrectly answered due to prompt context, and vice versa. These measures expose the model's susceptibility to contextual changes and its limited robustness. We will elaborate on this concept in the final version of the paper.
>
> - **Question E:** When referring to "first-word sampling," we intend to convey that the model tends to generate initial Yes/No responses, followed by elaboration through the reasoning process. This strategic approach prevents autoregressive models from influencing their yes/no predictions based on the reasoning process, thus preserving performance. We will rephrase this concept more explicitly in the upcoming version. Additionally, we would like to highlight the paper titled, [How Language Model Hallucinations Can Snowball
> ](https://arxiv.org/abs/2305.13534), which extensively delves into this phenomenon, offering an in-depth exploration of its intricacies.
>
> - **Question F:**  Drawing from our annotation process, we noted that even when provided with the capability to perform web searches, most human annotators refrain from extensive searches for specific volume parameters of objects. Instead, they depend on fragmented information to build foundational understanding. For instance, they might deduce that 'pollen' is a type of cell without necessarily looking up its exact size. Alternatively, they may search for images of the object, which may lack information about relative sizes. Hence, while external information enhances comprehension, it does not necessarily enable accurate mental comparison of object sizes.
>
> **Missing References:**
>
> We value your attention to related work and acknowledge the prior contribution by Hessam et al. We will certainly incorporate the reference and provide a discussion of their study within our revised submission to ensure due recognition of similar research in the domain.
>
> **Improvements and Corrections:**
>
> We extend our sincere gratitude for your meticulous review of grammatical, typographical, and presentational aspects.

---

### Official Review · Reviewer_GsC7 · 2023-08-03

**Soundness:** 4

**Excitement:**

4: Strong: This paper deepens the understanding of some phenomenon or lowers the barriers to an existing research direction.

**Missing References:**

None notable.

**Paper Topic And Main Contributions:**

The authors provide a new dataset for measuring LLM world knowledge, particularly about the comparative size of objects. The new dataset caters to important research questions surrounding in-context learning of LLMs. In particular, the controlled topic facilities provision of *no, partial, full*, and *incorrect* information in the LLM prompt. Results on a wide variety of LLMs (small, large, and commercial) provide interesting insights such as on gaps between human and LLM performance and alignment of human and LLM reasoning processes.

**Questions For The Authors:**

A: I am mostly interested in missing analysis I mention in the Reasons to Reject. Shedding light on these topics would be appreciated.

B: Does other human response data exist on this topic (i.e., from other studies)? Judging by the ample psycholinguistic backing for size comparison studies that is provided, one might expect there are existing statistics that may be informative to the reader.

**Reasons To Accept:**

The dataset is interesting, new, and well-designed. On the last point, its unique topic fills an apparent gap in the literature, allowing easily interpretable experimental design for the paper's main research questions on in-context learning. The takeaways are mostly clear, not well-known, and interesting. Writing is mostly clear.

**Reasons To Reject:**

**Main Point**
Sensitivity analysis on prompts and parameters is not provided. For instance, temperature can be an important factor in generating factual responses (as in the studied case). Similarly, LLMs can be sensitive to prompts in a way that is decidedly not human-like. Experimenting with a variety of different prompt formats / parameters and testing for consistency is an important, missing analysis. Doing so on the entire dataset, with all LLMs, and all prompts would be challenging - but it is reasonable to check these things on a smaller subset.

**Similar Minor Points**
 - Similarly to the above, but a more minor point, some sensitivity analysis on the answer mapping would be nice. For instance, your error analysis in the appendix indicates trouble with the "AIDS virus" - this may be caused by AI safety features.

 - There appears to be a general lack of analysis on statistical significance, which can be important when interpreting many results. This may cause the authors/readers to draw false positive inferences. I do consider this a minor point, since most takeaways are not inferred by "close calls" - most discussed differences in accuracy, for instance, are substantial.

 - Some results may be "over stated", for instance, the gap between ChatGPT performance and average human performance is < 20%. Arguably, this is the type of variability one might expect from humans of varying experiences/education levels.

*Post-rebuttal Revisions*
The main points are addressed by authors during the rebuttal, and authors have agreed to include these in the paper (Appendix with pointers in the main text is fine).

**Reproducibility:**

3: Could reproduce the results with some difficulty. The settings of parameters are underspecified or subjectively determined; the training/evaluation data are not widely available.

**Reviewer Confidence:**

3: Pretty sure, but there's a chance I missed something. Although I have a good feel for this area in general, I did not carefully check the paper's details, e.g., the math, experimental design, or novelty.

**Typos Grammar Style And Presentation Improvements:**

Typo in opening line 042.

Specific examples of prompts should be provided, this is not available in all cases.

---

> ### Author Rebuttal · Authors · 2023-08-29
>
> We extend our sincere gratitude for the dedicated time and meticulous evaluation provided by the reviewer. The insightful feedback is greatly valued, and we are eager to address the reviewer's points while elaborating on our approach and intentions:
>
> **Addressing the Concerns:**
>
> - **Sensitivity Analysis:** We agree with you that temperature is an important factor in the generation of LLMs. To alleviate the randomness effect of temperature, in all our experiments, we set the temperature as 0, so that the LLMs become deterministic. We agree that conducting the prompt and parameter tuning on a smaller subset will be beneficial, and we would like to provide the details about our existing tuning results here:
>
> 1. **GPT3-Davinci using Different Temperature Parameters** To investigate the effect of different temperature parameters, we conducted experiments on three different temperatures (0, 0.5, 1) on 100 randomly sampled test cases (50 questions for each type). The results show that setting the temperature to 0 can give the best results.
>
> | **Question** | **Prompt**            | **Accuracy** | **Macro-F1** | **Accuracy** | **Macro-F1** | **Accuracy** | **Macro-F1** |
> |--------------|-----------------------|--------------|--------------|--------------|--------------|--------------|--------------|
> | Temperature  |                       | **0**        |              | **0.5**      |              | **1**        |              |
> | General      | Plain Question        | 0.52         | 0.38         | 0.52         | 0.38         | 0.52         | 0.38         |
> |              | + Gold Size           | 0.78         | 0.78         | 0.80         | 0.80         | 0.78         | 0.78         |
> |              | + General Knowledge   | 0.50         | 0.33         | 0.50         | 0.33         | 0.50         | 0.33         |
> |              | + Size Knowledge      | 0.62         | 0.56         | 0.62         | 0.56         | 0.60         | 0.52         |
> | Special      | Plain Question        | 0.84         | 0.83         | 0.82         | 0.80         | 0.82         | 0.81         |
> |              | + Gold Size           | 0.94         | 0.93         | 0.90         | 0.90         | 0.92         | 0.92         |
> |              | + General Knowledge   | 0.84         | 0.81         | 0.86         | 0.83         | 0.82         | 0.78         |
> |              | + Size Knowledge      | 0.88         | 0.86         | 0.82         | 0.78         | 0.80         | 0.75         |
>
> **Table:** The results of GPT3-Davinci using different temperatures in different prompts on 100 randomly chosen test cases (50 questions for each type).
>
> 2. **GPT3-Davinci using Gold Size Information Prompt Templates** Since LLMs are sensitive to the format of the prompt, we use different prompt templates to probe their influence. We mainly test the gold size information prompt template and we list all templates in the Table.
> | **Template 1:** | The size of *Entity A* is *Exact Size of Entity A*. The size of *Entity B* is *Exact Size of Entity B*. + *Question* |
> |-----------------|--------------------------------------------------------|
> | **Template 2:** | *Entity A*: *Exact Size of Entity A*; *Entity B*: *Exact Size of Entity B*. + *Question*.  |
> | **Template 3:** | *Entity A* is *Exact Size of Entity A* and *Entity B* is *Exact Size of Entity B*. + *Question* |
> | **Template 4:** | *Question* + The size of *Entity A* is *Exact Size of Entity A*. The size of *Entity B* is *Exact Size of Entity B*. |
>
> We conduct experiments on the same 100 randomly sampled test cases (50 questions for each type). The effect of using different templates is notable on general questions and using Template 3 in the Table above has the best performance. Interestingly, Template 4 decreases the performance on both general and special questions, which means the order of the question and context is significant when choosing the prompt.
>
> | **Question** | **Prompt** | **Accuracy** | **Macro-F1** |
> |--------------|------------|--------------|--------------|
> | General      | Plain Question | 0.52 | 0.38 |
> |              | + Gold Size Information using Template 1 | 0.78 | 0.78 |
> |              | + Gold Size Information using Template 2 | 0.82 | 0.82 |
> |              | + Gold Size Information using Template 3 | 0.88 | 0.88 |
> |              | + Gold Size Information using Template 4 | 0.50 | 0.37 |
> | Special      | Plain Question | 0.84 | 0.83 |
> |              | + Gold Size Information using Template 1 | 0.94 | 0.93 |
> |              | + Gold Size Information using Template 2 | 0.94 | 0.95 |
> |              | + Gold Size Information using Template 3 | 0.90 | 0.89 |
> |              | + Gold Size Information using Template 4 | 0.86 | 0.84 |
>
> - **Answer Mapping Sensitivity:** We appreciate the reviewer's recommendation to explore the sensitivity of answer mapping, specifically in the context of AI safety mechanisms. We also find the intersection of AI safety and the physical-world understanding of LLMs to be a promising direction for our subsequent research.
>
> - **Statistical Significance and Overstatement:** We acknowledge the reviewer's valid concern regarding the necessity of conducting an extensive statistical significance analysis to reinforce the robustness of our findings. We will present the comprehensive significance test results for all of our key claims in the appendix of the final version.
>
>
> **Variability of Human Annotators:** While we acknowledge that variability of <20% could arise due to varying educational backgrounds among human annotators, our primary aim is to emphasise the comparison of object sizes as a foundational and direct task that mirrors LLMs’ grasp of the tangible world. Discrepancies in this aspect can potentially have significant implications in future interactions between Embodied LLMs and the physical reality. Thus, we emphasise the importance of recognizing the existing gap between human and LLM comprehension of the physical world, urging a thorough assessment of the present state of LLMs.
>
> **Responding to Questions:**
>
> - **Question A:** We deeply value the reviewer's attention to the analysis gap highlighted in the "Reasons to Reject'' section. We will incorporate the provided clarifications into the final version of the paper.
> - **Question B:** The reviewer's inquiry into the availability of human response data is well-noted. We also recognize the potential value in incorporating statistics from relevant studies.

---

### Official Review · Reviewer_gnhd · 2023-08-04

**Soundness:** 4

**Excitement:**

4: Strong: This paper deepens the understanding of some phenomenon or lowers the barriers to an existing research direction.

**Paper Topic And Main Contributions:**

The paper proposes a dataset (POSQA) to probe the real-world understanding of LLMs using size comparison questions. Authors show that LLMs struggle in zero-shot scenarios. They further investigate whether this understanding relies upon the context in the prompt or the internal weights of the model. They also study the impact of prompt formats and biases toward different objects. Their results indicate that LLMs struggle with the real-world understanding of different object sizes. Moreover, they have a preference for using the information in context over the knowledge stored in the weights. The results also suggest that LLMs' real-world understanding can be underestimated or overestimated depending on the prompt structure, affecting alignment with human behavior.

**Questions For The Authors:**

Give more details about the human annotation process used for the dataset.

**Reasons To Accept:**

1. Interesting, simple, and focused (size comparison) approach to probe real-world understanding in LLMs. The POSQA dataset is the main contribution of the paper.
2. The paper lays out the methodology and evaluation metrics in detail to systematically test the LLMs.
3. The results are relevant and interesting. They show that LLMs are still far from humans in real-world understanding. Moreover, model’s preference between context and weights is an important finding and can potentially lead to interesting future investigations.


**Reasons To Reject:**

1. The results might not be very general for actual “real-world understanding” of LLMs. Size comparison is still just one part of the bigger picture about real-world understanding.
2. The link to the dataset is not active. So, the claims made about the dataset can’t be verified. I would request the authors to look into it.
3. No details about the human annotation process used for comparison with ChatGPT.

**Reproducibility:**

2: Would be hard pressed to reproduce the results. The contribution depends on data that are simply not available outside the author's institution or consortium; not enough details are provided.

**Reviewer Confidence:**

3: Pretty sure, but there's a chance I missed something. Although I have a good feel for this area in general, I did not carefully check the paper's details, e.g., the math, experimental design, or novelty.

**Typos Grammar Style And Presentation Improvements:**

Line 392 - "It is surprised"
Line 89 - "our contributes can"

---

> ### Author Rebuttal · Authors · 2023-08-29
>
> We extend our sincere gratitude to the reviewer for conducting a comprehensive evaluation of our paper. We highly value the opportunity to address the reviewer's insights and provide further insights into our research. Below, we address the concerns raised by the reviewer and provide additional clarifications:
>
> **Reasons to Address the Concerns Raised by the Reviewer:**
>
> - **Scope of Real-World Understanding:** We acknowledge the reviewer's concern that our study's focus on size comparison represents only a facet of real-world understanding. We concur that real-world understanding is multifaceted, and our work serves as an exploratory step into a specific aspect. We intend to clearly contextualise this aspect within the broader understanding of language models' capabilities in both our introduction and conclusion.
>
> - **Dataset Accessibility:** We sincerely apologise for the inaccessibility of the dataset link in the original submission. To address this, we are delighted to provide access to the [preview version of POSQA](https://anonymous.4open.science/r/POSQA-EF54/datasets). Upon acceptance, we commit to releasing the dataset on the Hugging Face Datasets platform to facilitate widespread usage and validation.
>
> - **Human Annotation Process:** We greatly appreciate the reviewer's interest in our human annotation process for ChatGPT comparison. We are pleased to provide detailed information about the [online](https://anonymous.4open.science/r/POSQA-EF54/annotation_online) and [offline](https://anonymous.4open.science/r/POSQA-EF54/annotation_offline) annotations, including text excerpts, labelled categories, and annotation durations. Our annotators, all with master's degrees and advanced English proficiency, utilised the [Potato Platform](https://potato-annotation-tutorial.readthedocs.io/en/latest/) for annotation. In our final version, we will comprehensively outline the annotation process in the appendix.
>
> **Addressing Typos, Grammar, and Presentation:**
>
> We express our gratitude to the reviewer for their meticulous attention to detail. We will diligently rectify the typographical errors mentioned, and aim to augment the clarity
>
> **Reproducibility and Transparency:**
>
> We acknowledge the reviewer's concern regarding the reproducibility of our results. In response, our revised submission will provide an in-depth exposition of our methodology, experimental design, and evaluation metrics. Moreover, we will offer a comprehensive overview of the dataset's structure and components, ensuring accessibility for external researchers to comprehend and replicate our approach.
>
> **Future Directions and Contributions:**
>
> We deeply value the reviewer's recognition of our study's contribution to comprehending LLMs' real-world understanding. We concur with the reviewer that our findings open avenues for investigating the intricate interplay between context and model weights. In our final version, we will elaborate on potential research directions within this domain, fostering further exploration and collaboration within the research community.

---

### Meta-Review · Area_Chair_pMGX · 2023-09-23

**Recommendation:** 3

**Metareview:**

The mojority decision of the reviewers does a fair job of weighting the pros and cons of the work as presented. The updated results and cleanup of the material should be reflected in further edits of the paper inorder to clarify the discussion and highlight the contributions.

---

### Decision · Program_Chairs · 2023-10-07

**Decision:**

Accept-Findings

**Comment:**

The mojority decision of the reviewers does a fair job of weighting the pros and cons of the work as presented. The updated results and cleanup of the material should be reflected in further edits of the paper inorder to clarify the discussion and highlight the contributions.